# Mycosporine-Like Amino Acids (MAAs) in Time-Series of Lichen Specimens from Natural History Collections

**DOI:** 10.3390/molecules24061070

**Published:** 2019-03-19

**Authors:** Marylène Chollet-Krugler, Thi Thu Tram Nguyen, Aurelie Sauvager, Holger Thüs, Joël Boustie

**Affiliations:** 1CNRS, ISCR (Institut des Sciences Chimiques de Rennes)-UMR 6226, Univ Rennes, F-35000 Rennes, France; marylene.chollet@univ-rennes1.fr (M.C.-K.); ntttram@ctump.edu.vn (T.T.T.N.); aurelie.sauvager@univ-rennes1.fr (A.S.); 2Department of Chemistry, Faculty of Science, Can Tho University of Medicine and Pharmacy, 179 Nguyen Van Cu Street, An Khanh, Ninh Kieu, Can Tho, 902495 Vietnam; 3State Museum of Natural History Stuttgart, Rosenstein 1, 70191 Stuttgart, Germany; 4The Natural History Museum London, Cromwell Rd, Kensington, London SW7 5BD, UK

**Keywords:** herbarium, fungarium, mycosporine-like amino acids, degradation, storage, *Dermatocarpon luridum*

## Abstract

Mycosporine-like amino acids (MAAs) were quantified in fresh and preserved material of the chlorolichen *Dermatocarpon*
*luridum* var. *luridum* (Verrucariaceae/Ascomycota). The analyzed samples represented a time-series of over 150 years. An HPLC coupled with a diode array detector (HPLC-DAD) in hydrophilic interaction liquid chromatography (HILIC) mode method was developed and validated for the quantitative determination of MAAs. We found evidence for substance specific differences in the quality of preservation of two MAAs (mycosporine glutamicol, mycosporine glutaminol) in Natural History Collections. We found no change in average mycosporine glutamicol concentrations over time. Mycosporine glutaminol concentrations instead decreased rapidly with no trace of this substance detectable in collections older than nine years. Our data predict that a screening for MAAs in organism samples from Natural History Collections can deliver results that are comparable to those obtained from fresh collections only for some MAAs (e.g., mycosporine glutamicol). For other MAAs, misleading, biased, or even false negative results will occur as a result of the storage sensitivity of substances such as mycosporine glutaminol. Our study demonstrates the value of pilot studies with time-series based on model taxa with a rich representation in the Natural History Collections.

## 1. Introduction

Natural History Collections provide a rich source of preserved material for the study of organisms, which otherwise could only be accessed with an excessive logistic effort [1]. For species that are now extinct in the wild or where natural populations are inaccessible because of political unrest and war or natural disasters and thus cannot be cultured, access to Natural History collections is the only remaining choice [2]. At the same time, these specimens are priority parts of the cultural heritage and the potential outcome of any invasive sampling needs to be balanced against the impact of the sampling method on the preservation of representative parts of the museum specimens [3]. Invasive sampling can be mandatory, even for routine identification in some organism groups, for example, for lichenized fungi, the need for the study of anatomical characters and chemotaxonomical markers as part of routine identification is widely accepted. For the taxonomy of lichenized fungi, having characters based on the visualization of characteristic metabolites or their reaction with selected test reagents has a long tradition. Already in the 19th century, Nylander introduced so-called spot-tests in lichen taxonomy and species identification checks for taxon-specific colour changes of lichen tissue following the application of solutions of calcium hypochlorite and potassium hydroxide, which are still in use as routine elements in lichen identification [4]. In the 20th century, the visualization of metabolite profiles by thin layer chromatography has become an additional routine tool for chemotaxonomic purposes [5]. More advanced techniques such as HPLC-DAD have mostly been used only in collaborative projects between lichen taxonomists and specialized natural product chemists [6]. The much smaller amount of material needed for analyses with the latest equipment is constantly improving the balance between the gain of new scientific data and the loss of valuable material as a result of invasive sampling procedures. The analysis of metabolite profiles is also the basis for studies in the interaction of lichens with their environment, for example, their palatability for herbivores [7,8], for the penetration of UV-light into the thallus [9], the complexation of metal ions [10,11], and the hydrophobicity of lichen surfaces [12]. 

In recent years, lichens have become increasingly popular among natural product researchers as a source of a large diversity of natural products [13]. Some of these substances provide human health benefits, crude preparations of some lichens are traditionally used as components of condiments, dyestuff, and in folk medicine, and they played an important role in the perfumery industry. 

Lichens cannot be cultured on a large scale and harvesting of wild populations for food or medical purposes has become a serious threat for some overexploited species [14]. Limited access to wild populations and an increasing need for new lead chemicals in pharmaceutical research has renewed interest in the screening of lichens for novel metabolites, including from material stored in Natural History Collections [15]. 

For studies in chemotaxonomy, however, data from herbarium collections can only be compared to those from fresh collections, if the substances under study can be shown to be unaffected by storage time and environmental conditions typical for herbaria, for example, the presence of volatile chemicals from slowly degrading labels or capsules due to non-archival quality materials that may contain acidic components. Much effort has been invested in the study of agents of deterioration for the DNA in biological material from Natural History Collections [2,16], but very few studies have addressed the stability of secondary metabolites in collections over time [17].

Analytical research on lichen metabolites has long focused on acetone soluble substances, partly because of the widespread acceptance of the standard protocol by Culberson & Christinsson using thin layer-chromatography as a routine technique in lichen identification via the visualization of characteristic metabolite patterns [5]. The procedure proposed in this standard protocol focused on acetone-soluble metabolites such as depsides, depsidones, and fatty acids, because these include the most widespread and diagnostically significant substances in most of the lichen forming fungi. Instead, water-soluble substances and their use as chemotaxonomic markers were largely neglected in the routine practice of most lichen taxonomists. This includes mycosporines and mycosporine-like amino acids (MAAs), which were only recently explored in lichens [18,19].

Mycosporine-like amino-acids (MAAs) are small polar metabolites composed of a cyclohexenone or cyclohexenimine ring structure conjugated to an amino acid or an amino alcohol group. They are characterized by a unique strong absorption peak in the 310–360 nm UV range with a λmax around 310 nm for MAAs with a cyclohexenone ring. To date, at least 50 MAAs and their glycosylated derivatives have been described. In living organisms, they are multifunctional molecules with effects on resistance to ultraviolet radiation intensity, osmotic stress, desiccation, temperature extremes, and antioxidants, along with a possible role as a nitrogen reservoir [20,21]. They have been isolated from various pro- and eukaryotic organism groups, for example, cyanobacteria, corals, fish, and fungi, but not from higher plants and are known from various eukaryotic algal phyla [22,23,24], where taxon-specific differences in the adaptability of MAA contents to environmental conditions have been observed [25]. In lichens, mycosporines were long believed to be restricted to thalli, which contain lichen forming fungi associated with cyanobacterial photobionts. Only recently, two MAAs, mycosporine glutaminol (**1**) and mycosporine glutamicol (**2**) (Figure 1), were also detected in the chlorolichens *Dermatocarpon luridum* and *D. miniatum* (Verrucariaceae, Ascomycota) [26], two species associated with the eukaryotic alga *Diplosphaera chodatii* (Prasiolaceae, Chlorophyta) [27].

*Dermatocarpon miniatum* auct. is a species complex with several morphologically identical lineages that can only be separated by DNA-barcoding, which makes an unambiguous identification of old herbarium collections difficult [28]. 

*Dermatocarpon luridum* and its associated photobionts have been studied genetically in some detail [29,30]. While North-American populations can be difficult to separate from similar species by morphological characters and simple chemical “spot tests”, for example, the colour change of the medullary layer after application of Meltzer’s reagent [31], these difficulties were not known from European populations of the typical variant [32]. Genetic diversity of both myco- and photobiont in European lichens formed by *Dermatocarpon luridum* var. *luridum* and its associated algae *Diploshaera chodatii* is surprisingly low, even between geographically distant populations [29]. This lichen forms large colonies on seeping rocks and in freshwater streams [33]. While assessing collections at the Natural History Museum London (BM) and Rennes1 University Herbarium (REN), we found that among all Verrucariaceae, this is one of the taxa with the best representation across geographic range and time. Collections were large enough and of unambiguous identity to allow sampling of sufficient quantities of thallus for MAA analysis without leaving a visible trace of the invasive sampling on the collection. The typical variant of this species was thus chosen as the model taxon for our study. 

Our aim was to study the influence of storage time on the content of two different mycosporines (mycosporine–glutamicol and mycosporine–glutaminol), which were known to occur in fresh collections of *Dermatocarpon luridum* [26]. Mass spectrometry techniques such as direct analysis in real time (DART) or laser-desorption ionization (LDI) were previously used to detect the presence of mycosporine serinol from very small amounts of the lichen thallus *Lichina pygmaea* [34,35]. Technically these methods are applicable to herbarium specimens as well, but we preferred to develop a simpler protocol allowing the quantification of these mycosporines. Capillary electrophoresis has recently been described as a method for the analysis of MAAs, but it requires a specific device and is not a widespread technique because of its limitation to ionisable compounds [36]. So far, the common quantification method used was HPLC-DAD with a reverse phase C18 column [37]. Hartmann et al. recently published an alternative approach by hydrophilic interaction liquid chromatography (HILIC) to improve access to the concentration of such metabolites [38]. C18 was replaced by bare silica particles, a highly polar stationary phase in HILIC mode, which enables the separation of highly polar compounds. In this context, an HPLC-DAD in HILIC mode using an internal calibration curve with cytosine as an internal standard has been developed for mycosporines’ quantification. The validation method was performed with respect to linearity, precision, limit of quantification (LOQ), and limit of detection (LOD). The results of our study provide vital data for answering the question to which extend older herbarium collections can help to complement sampling gaps of fresh lichens for mycosporines in studies on chemical ecology and for further screening of lichenized fungi and other organism groups preserved in the Natural History Collections for novel forms of MAAs.

## 2. Results

### 2.1. Validation of Analytical Procedure

Linearity was validated by performing six different concentrations of mycosporine standards along with the internal standard cytosine in triplicate. The corrected peak area and concentration were subjected to the regression analysis to calculate the calibration equation and correlation coefficients. Satisfactory correlation coefficients (*R*^2^ > 0.992) were obtained within a calibration range of 5.00–40.00 µg/mL for **1** and **2** (Table 1). The precision of the analytical method, investigated by performing six consecutive replicate injections of the same standard solution, was within accepted limits—4.21% and 2.27% for **1** and **2**, respectively. The LODs and LOQs were determined as 1.41 µg/mL and 2.93 µg/mL, respectively, for **1**, and 4.49 µg/mL and 8.99 µg/mL, respectively, for **2**.

### 2.2. Analysis of Lichen Collections

HPLC separation of aqueous extracts indicated the presence of two mycosporines **1** and **2** based on matching retention times (5.07 min and 5.71 min, respectively) and their characteristic UV-spectra (Figure 2). We did not find signs of further MAAs in the tested material. The HILIC-HPLC-DAD with a total run time of 16 min appeared to be a good choice for the study of numerous samples.

Quantification of mycosporines glutaminol and glutamicol was calculated using the calibration curves of the respective standards. Mycosporine glutamicol was found in 45 out of 48 collections of *Dermatocarpon luridum* var. *luridum*. Only two collections from Western Scotland (collected in 1905 and 1959) and one from the river Teign in SW England did not show any trace of MAAs. Other collections from the same area and with similar storage time did contain mycosporine glutamicol well above the detection limit (but not mycosporine glutaminol), and we have not found any further geographic pattern for the variation in mycosporine glutamicol concentrations. Mycosporine glutamicol contents varied between 4.56 and 34.89 mg/g of dried extract, corresponding to 0.15–1.12 mg/g of dried lichen (Figure 3), with the four highest concentrations being recorded for samples collected in the years 2006, 1933, 1886, and 1864. In six samples, the distinctive absorption pattern of mycosporine glutamicol was clearly visible and was treated as a qualitative signal for the presence of mycosporine glutamicol, but remained too low for accurate quantification. Average mycosporine glutamicol concentrations among the sampled collections remained constant, irrespective of the time of collection (e.g., from 1861 to 2013). Mycosporine glutaminol was found in 7 out of 48 collections of *Dermatocarpon luridum* var. *luridum*. Measured concentrations varied between 2.82 and 15.62 mg/g of dried extract, corresponding to 0.09–0.50 mg/g of dried lichen. All collections with detectable mycosporine glutaminol concentrations were less than ten years old. 

## 3. Discussion

Although mycosporine-like amino acids (MAAs) have attracted a lot of attention by ecologists [39], pharmacologists [20], and even materials science researchers [40], their structural diversity, evolution, and distribution among extant species is still poorly understood and published data are often based on small numbers of specimens. 

Our data largely confirm earlier observations [26] of the presence of only two MAAs, mycosporine glutamicol and mycosporine glutaminol, in the thalli of *Dermatocarpon luridum* var. *luridum*. The complete lack of detectable quantities of MAAs in two collections of this taxon from different localities in Western Scotland and one in SW England was unexpected. Whether there are ecological conditions that reduce MAA-concentrations and whether there are undescribed cryptic taxa in this area remains to be clarified. Our starting assumption of a low genetic diversity in European *D. luridum* var. *luridum* was based on previous analyses with a strong sampling bias towards populations from Central Europe [28], and although we aimed to minimize the risk of including representatives of undescribed cryptic look-alikes by choosing a model taxon for which no cryptic infraspecific differentiation or unrelated look-alikes were known, the existence of such taxa cannot be ruled out entirely. There are also no data yet that would allow an assessment of which ecological variables influence MAA-concentrations in lichens of the family Verrucariaceae.

Experiments on the stability of MAAs under laboratory conditions have shown substance-specific differences in the stability under different pH and temperature conditions and concluded that predictions of stability to certain agents of deterioration for MAAs based on their structure are currently still unreliable [20]. Furthermore, all experiments on MAA stability have tested only short-term exposure and rather extreme pH and temperature conditions [41,42,43] that are unlikely to occur in well-curated Natural History Collections. To our knowledge, no study so far has aimed to test the long-term stability of MAAs (e.g., decades) under room temperatures and address its relevance for the interpretation of results of screening programs using older Natural History Collections. 

The analysis of our time-series shows a surprisingly strong difference in the preservation of the two MAAs over time in our model taxon. Concentrations of mycosporine glutamicol are apparently unaffected by storage time, but mycosporine glutaminol quickly disappears from stored material, within less than a decade. The products of the mycosporine glutaminol degradation are still unknown in lichen thalli, however, Pittet and Bernillon have demonstrated that under a slightly acidic treatment during a cationic exchange chromatography, compound **1** is easily hydrolyzed into compound **2** [44,45]. It remains to be tested if this process also occurs in Natural History Collections, for example, by repeat measurements of the samples tested in our study after 5 to 10 years of additional storage time. 

In order to prevent the degradation of acidity sensitive substances (including DNA) in Natural History Collections, curatorial protocols emphasize the importance of acidity-free and buffered materials for the paper used in herbarium sheets or storage capsules [46]. Modern curatorial practice in large Natural History Collections, for example, at the BM collection, does take this aspect into account and the use of acid-free capsules, annotation slips, and labels became standard more than 30 years ago. Nevertheless, mycosporine glutaminol concentrations still declined in packets from the BM collection in the same way as they did in collections elsewhere and the avoidance of acidic materials in direct contact to the lichen appears to be insufficient to protect sensitive MAAs from degradation. Other sources of acidic substances could be volatile organic acids produced from the decay of older, non-archival quality packet and herbarium sheet materials in historic collections housed in the same room as the more recently collected specimens. Small amounts of volatile organic acids also develop already within a few months from spontaneous oxidative decay of cellulose even in paper branded as acid-free or with alkaline filling substances [47].

Our results have a direct implication for the use of herbarium samples in chemotaxonomic studies, studies on character evolution, chemical ecology, or screening programs for novel MAAs. For such study types, a taxon sampling of maximum width across the studied group is essential for the assembly of a robust and meaningful dataset. Fresh collections will always be the first choice for studies on natural products, but they can be difficult to obtain for naturally rare or threatened species or for populations in areas that are inaccessible and in cases where the habitat of the target species no longer exists [1,3]. In such cases, preserved samples are highly desirable sources for metabolite analysis. Our study was initiated as a prerequisite test in preparation for a large-scale screening for MAAs across the species rich lichen family Verrucariaceae, including those taxa that, because of logistical reasons, are difficult to obtain freshly from the wild.

Our data show for the first time, that MAAs are detectable in Natural History Collections, but for a given compound, the time of storage can lead to its full disappearance. With this limitation in mind, it would be questionable to include specimens that are older than at most 10 years in general MAA screening programs if they are represented in Natural History Collections with only small amounts of biomass and if they represent very rare or extinct species. Recently collected (<10 years from the time of analysis) material instead can be valuable for qualitative screening (chemotaxonomy) and for the search of taxa with potentially new MAA-candidates, but it is important to keep in mind that the ratio between MAAs may be affected by storage effects as soon as a few years after collection. Applications of data obtained from herbarium collections, for example, for inclusion in studies on chemical ecology in which data on concentrations rather than simple absence/presence may be required, are only possible when material of a similar age is compared and should generally be seen as a last resort.

Substance-specific differences in the long-term stability of MAAs directly affect their use in potentially long living technical applications, for example, as UV-filters in sun-screen panels [40]. Our results show that degradation of MAAs may not be visible in short-term exposure experiments, but can lead to total loss of UV-absorbance within a couple of years if MAAs that have not been tested in longer time-series studies are used.

## 4. Materials and Methods 

### 4.1. Specimen Selection

We have selected a total of 48 specimens from the Natural History Collections at BM and REN together with fresh collections by the authors and recently collected material obtained from members of the French Association of Lichenology. These samples represented a time-series of European samples of *Dermatocarpon luridum* var. *luridum* from the mid-19th century to the present day (Appendix A). Specimens were chosen for MAA identification and quantification—whether a check of the identification confirmed the correct identity of the lichenized fungus and whether the thalli were free of excessive sediment deposits. For the late 19th century specimens from the private collections of H.B. Holl, collection dates are missing on the labels, but the time of the transfer of his collection to the public herbarium at the Natural History Museum London (BM) after his death in 1886 [48] is documented. In these cases, the calculated number of years of storage represents the minimal length of time and may be several years lower than the time since the real date of collection.

### 4.2. Sample Preparation

First, 15 mg of uncrushed lichen material, corresponding to about 1 cm^2^ of dried *Dermatocarpon luridum* thallus, was extracted with 300 µL of water at 4 °C for 2 h. The supernatant was filtered (0.45 µm) and lyophilized to dryness, the yield of the extraction was about 3.2%. Crude aq. extracts were then dissolved in water and internal standard (IS) solution to give a final concentration at 1 mg/mL. Two repeated measurements of each sample solution were analyzed by HPLC.

### 4.3. Method Validation

Mycosporine glutaminol (**1**) and mycosporine glutamicol (**2**) isolated from *D. luridum* [26] were used as standards. Their structures were determined by ESI+-HRMS, NMR, and IR spectra, as already published as Appendix A in the work of [26]. In order to perform calibration curves, stock solutions in water of standards **1** and **2** (1 mg/mL) and the internal standard (IS) cytosine (250 µg/mL) were prepared. A wide range of six calibration standards (5–40 µg/mL) including IS (25 µg/mL) were prepared in triplicate by suitable dilution in water. Calibration curves were obtained by plotting the corrected area (ratio: mycosporine area at 310 nm/IS area at 254 nm) for each standard solution versus the corresponding mycosporine concentration. The precision was investigated by performing six consecutive replicate injections of the same standard solution and the peak area measured was expressed in terms of the percentage of relative standard deviation. LOD and LOQ were calculated following the formulas LOD = (b + 3 σ(b)/a) and LOQ = (b + 10 σ(b)/a) with a: slope, b: intercept of calibration curve, and σ(b): standard deviation of b.

### 4.4. HILIC-HPLC-DAD Analysis

Each sample solution and the mycosporine standards (10 µL) were injected into an HPLC-DAD system (Shimadzu^®^, Marne La Vallée, France, LC-20 AD SP, injector SIL-20A HI, column oven CTO-20A, diode array detector SPD-M20A) using a Kinetex HILIC 100 Å (2.6 µm, 100 × 4.60 mm) column, with mobile phase A (ACN/CH_3_COONH_4_ 50 mM 90:10, pH 5.36) and mobile phase B (ACN/H_2_O/CH_3_COONH_4_ 50 mM 50:40:10, pH 5.36). The elution followed the following gradient: 100% of A 0–2 min, 0%–100% of B 2–4 min, 100% of B 4–12 min, 0%–100% of A 12–14 min, 100% of A 14–16 min. The flow rate and column temperature were set to 1 mL/min and 40 °C, respectively. Peak detection was carried out online using a diode array detector at 254, 310, 330, and 360 nm, and absorption spectra (200–400 nm) were recorded each second directly on the HPLC. Compounds **1** and **2** in extracts were identified by comparison of their retention time, 5.07 min for **1** and 5.71 min for **2**, with those of reference standards at 310 nm. Cytosine as IS was detected at 254 nm with a retention time = 2.50 min. 

## 5. Conclusions

Screening samples from preserved Natural History Collections for mycosporine-like amino acids is a promising approach to overcome logistical challenges when samples have to be included from across the world or from sites that have been altered to the extent that re-collecting the target species is no longer possible. Reliable results, however, can only be expected from substances that are resilient to decay in the collection environment. This stability is shown here to be the case for mycosporine glutamicol, but not for mycosporine glutaminol. While during the first decade of storage, at least qualitative data (presence/absence) may be obtained for both mycosporine glutamicol and mycosporine glutaminol, quantitative data obtained from preserved specimens have to be considered as unreliable already after a few years in storage as a result of the decrease of mycosporine glutaminol concentrations.

Our case study demonstrates the importance of pilot studies to check the storage sensitivity of the targeted MAA types with time-series of collections in carefully selected model taxa—avoiding species or varieties that are difficult to identify in old collections and without the need for DNA-barcoding—before the sacrifice of material from particularly rare species or valuable historical specimens can be justified in upscaled screening programs.

However, this study reinforces the interest of scarce studies stressing the value of herbarium material, including lichens [49], as the stability of some secondary metabolites is found to be poor or not sensible to time degradation [50]. This study focused on a special metabolite pattern with a dedicated analytical approach, highlighting the presence of these compounds but limiting observation of other metabolites. The accuracy and efficacy of analytical tools are constantly improved, allowing for analysis with highly informative data from micro-sampling. Non-destructive methods such as indirect LAESI-MS [51] and UPLC-HRMS-DADanalysis from a droplet extraction [52] were recently used. Such methods will certainly be optimized through molecular networking approaches to acquire large amounts of accurate data [53]. Therefore, herbaria samples and series collected from ancient times appear to represent an invaluable opportunity to obtain information about the stability of secondary metabolites. The quantity and variety of data obtained from respectful sampling from specimens in Natural History Collections will provide priceless information about the chemical diversity as announced by the term herbariomics [54]. Such information will find applications in chemotaxonomy, chemical ecology, and many other fields with the help of chemoinformatics. 

## Figures and Tables

**Figure 1 molecules-24-01070-f001:**
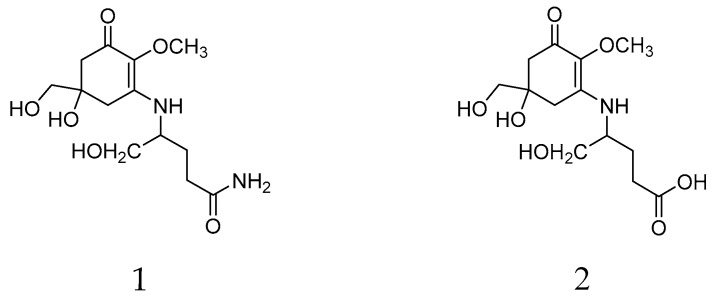
Chemicals structures of mycosporine glutaminol (**1**) and mycosporine glutamicol (**2**) isolated from *Dermatocarpon luridum* var. *luridum*.

**Figure 2 molecules-24-01070-f002:**
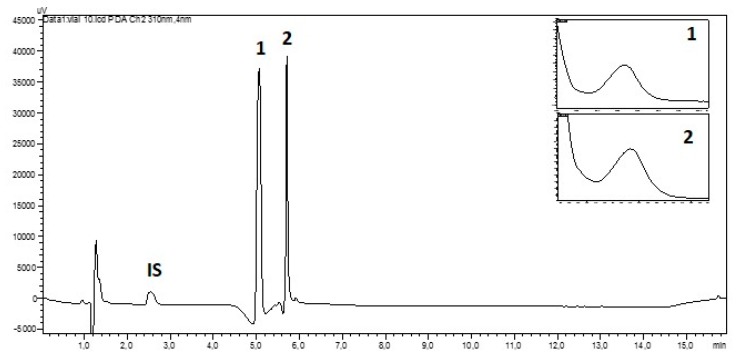
Hydrophilic interaction liquid chromatography (HILIC)-HPLC chromatogram of the aqueous extract from a sample of *Dermatocarpon luridum* var. *luridum* (voucher barcode *JB/001/07/2012*) recorded at 310 nm and inset UV-spectra of mycosporines **1** (Rt = 5.07 min) and **2** (Rt = 5.71 min) (**IS**: Internal Standard added to *D. luridum* extract, **x-axis**: 0–16 min, **y-axis**: arbitrary units of absorbance).

**Figure 3 molecules-24-01070-f003:**
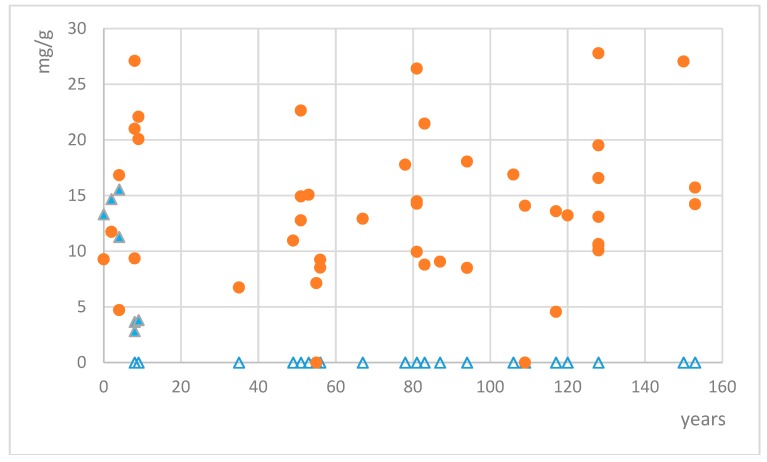
Concentrations of mycosporine glutamicol (red circles) and mycosporine glutaminol (blue triangles, empty triangles = no mycosporine glutaminol detected) in dried extracts of fresh and preserved European specimens of the lichen *Dermatocarpon luridum* var. *luridum*.

**Table 1 molecules-24-01070-t001:** Validation results of the hydrophilic interaction liquid chromatography (HILIC) method for mycosporines **1** and **2**. LOD—limit of detection; LOQ—limit of quantification.

	Mycosporine Glutaminol (1)	Mycosporine Glutamicol (2)
Concentration range (µg/mL)	5.00–40.00	5.00–40.00
Regression equation	y = 0.04x − 0.03	y = 0.03x + 0.09
Correlation coefficient (*R*^2^)	0.9984	0.9922
Precision (%)	4.21	2.27
LOD (µg/mL)	1.41	4.49
LOQ (µg/mL)	2.93	8.99

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
