# Peer review of "Mycosporine-Like Amino Acids (MAAs) in Time-Series of Lichen Specimens from Natural History Collections"

_molecules, 2019, doi:10.3390/molecules24061070_

Round 1

Reviewer 1 Report

The manuscript described the quantitative study of mycosporine glutamicol and mycosporine glutaminol in in fresh and preserved materials of the chlorolichen Dermatocarpon luridum var. luridum (Verrucariaceae/Ascomycota) using HPLC-DAD with HILIC mode method.  Mycosporine-like amino acids (MAAs) are widespread and have been reported in a wide variety of microorganisms including bacteria, fungi, algae, and mosses as well as higher plants, macroalgae and marine animals. There are many quantitative studies1 on MAAs. The method used in this manuscript is very similar to that published by Hartmann and colleagues2. Some MAAs may have the same UV and retention times, so the authors should at least use LC/MS method to determine the concentration of more MAAs, not just mycosporine glutamicol and mycosporine glutaminol.

1. R. PRZESLAWSKI,* K. BENKENDORFF, and A. R. DAVIS “A QUANTITATIVE SURVEY OF MYCOSPORINE-LIKE AMINO ACIDS (MAAS) IN INTERTIDAL EGG MASSES FROM TEMPERATE ROCKY SHORES” Journal of Chemical Ecology, 2005, Vol. 31, No. 10, 2417

2. Anja Hartmann, Adele Murauer, Markus Ganzera* “Quantitative analysis of mycosporine-like amino acids in marine algae by capillary electrophoresis with diode-array detection” Journal of Pharmaceutical and Biomedical Analysis 2017, 138, 153–157

Author Response

Thank you very much for these two relevant references. We already referred to the HILIC quantification of MAAs published by Hartmann et al in 2015 as we adapted this protocol for our study. We mentioned the capillary electrophoresis technique and added the corresponding reference to be included in our choice explanation given in the following sentence:

Line 118:“For quantification method, capillary electrophoresis has recently been described for the analysis of MAAs but it requires a specific device and is not a widespread technique as limited to ionisable compounds [36].”

Reference to the publication of Przeslawski 2005 is added line 121 as the new ref 37 to support our statement about a routine method.

We totally agree that some MAAs may have the same UV and retention times. That’s why our previously work (ref 26) has clearly demonstrated, by a HPLCDADMS2 analysis of an aqueous extract of the lichen Dermatocarpon luridum var. luridum, the presence and the identification of only two MAAs: mycosporine glutamicol and mycosporine glutaminol. These two MAAs have also been isolated as the only MAAs in many D. luridum samples and duly identified by spectrometric methods. So, they were used as standards for this new study. The goal of the present study carried on this lichen species was not to identify new MAAs but only to quantify these two MAAs in a time series of Dermatocarpon luridum specimens. To our knowledge, the best method for MAAs quantification is a LC/DAD method (thanks to a strong absorption at 310 nm) instead of LC/MS method which is very less sensible (small ionization in ESI+).

Reviewer 2 Report

The present manuscript is a report on small, mycosporine-like amino acids (MAAs) that were detected in lichens from Herbarium specimen ranging from recent to over 150 years in age. The focus is on two specific MAAs: mycosporine glutaminol and micosporine glutamicol. One of these, microspore glutaminol, was not detected in samples older than nine years. The other MAA, microspore glutamicol, was found in varying content in all but three tested samples.

A couple of remarks/questions:

Introduction line 86ff or discussion line 172: How many MAAs have been reported so far apart from the two described ones (overall, not only in Dermatocarpon luridum where only these two occur?

Line 78: MAAs are water soluble; can they be flushed off by heavy rain and could this be a reason why glutamicol was not detected in the samples from Western Scotland and SW-England, respectively (line 179-180)? Please comment.

And in accordance with this ecological aspect, do authors expect a stronger/faster loss of MAAs in the wild, i.e. by climatic changes due to winter/summer season or humid/dry conditions at the time of sampling (as compared to stable herbarium conditions; lines 194-195)?

Line 83: “… They are characterized by a unique strong absorption peak in the 310-360 nm UV range…”. However, only data for 310 nm are given in fig 2. What about the range/other wavelengths of 320-360?

Fig. 2: the units on x and y axes are too small and therefore difficult to read; please, include a legend.

Fig. 3: please, comment on the varying content of glutamicol (in the results section). In the figure legend, it should be (blue circles) instead of triangles for glutaminol and the lichen species name should be in italic.

Discussion line 208: Are there differences in the (historical) use of herbarium sheets (paper, capsules, envelopes, etc.) in the two Natural History Collections in London and in Rennes? Have the authors compared samples of similar age out of the two difference Collections?

Line 106: What is the meaning of the small "1" after Rennes?

Author Response

Introduction line 86ff or discussion line 172: How many MAAs have been reported so far apart from the two described ones (overall, not only in Dermatocarpon luridum where only these two occur?

From literature (ref 20, 21), we counted 52 MAAs described in water related organisms. We added the sentence:

Line 84: “To date, at least 50 MAAs and their glycosylated derivatives have been described.“

Line 78: MAAs are water soluble; can they be flushed off by heavy rain and could this be a reason why glutamicol was not detected in the samples from Western Scotland and SW-England, respectively (line 179-180)? Please comment.

This is a legitimate question to which unfortunately only speculative answers can be given at this point due to the lack of specific experiment to address this question. From W-Scotland we have analyzed a total of five collections, but only two of them lack measurable traces of MAAs.  Similar high-rainfall and eu-oceanic climatic conditions also apply to the specimens from Ireland and the Faroe-Islands, but all of these contained Mycosporine glutamicol in concentrations well above the detection limit.

The specimen from SW-England with no MAAs detected is from Dartmoor, a region within the county of Devon (we have made this clearer now in the supplementary table). We have analyzed nine further specimens from this county, all of them with Mycosporine glutamicol contents well above the detection limit. This part of Britain has a rather oceanic climate with frequent rainfall throughout the entire year.

We do not know yet where in the lichen thallus MAAs are stored – if the storage is intracellular the easy extraction would be surprising. If water soluble MAAS are kept within the intercellular space in the lichen thallus by a still unknown mechanism or if they are constantly produced in order to balance losses is unknown and certainly merits further research.

Dermatocarpon luridum var. luridum is generally a characteristic lichen of either temporarily inundated micro-sites or places which receive frequent spray from splash water. Wet conditions and even water saturation events are therefore not a weather dependent exception but a regular feature of its natural habitat, particularly in the west of the British Isles where regular water level fluctuations are generally low compared to more continental parts of Europe. If MAAs would be easily flushed off (e.g. by heavy rain), they should not occur in substantial amounts in this lichen at all, except after unusual drought periods. The labels of most of the historic collections however indicate that the specimens where inundated at the time of the collection. If MAAs would be easily washed out by naturally occurring water logged conditions samples which were taken from below the water surface should not have the high MAA concentrations which we did measure in the vast majority of samples.

Theoretically for those colonies which are not permanently submerged (Dermatocarpon luridum is typically best developed at micro-sites with temporary but not permanent inundation) it may be possible that MAAs are better retained in more or less continuously wet thalli, but that after unusual long dry periods with low water levels in the streams the retention or replenishing mechanism may need some time to be fully efficient and therefore under such weather conditions a single heavy rain may cause larger than usual losses of MAAs. However, such a scenario would be expected to occur much more often in more continental parts of the study area and would be least likely in the highly oceanic regions where we did detect specimens without measurable amounts of MAAs. Therefore we believe that the most likely explanation is, that MAA-deficient lineages of Dermatocarpon co-occur with typical D.luridum var. luridum in some sites along the British Western Coastline. If these lineages merit a formal taxonomic rank, or if they are occasional variations which survive more easily under highly oceanic climate remains to be studied.

Please also see our comment in lines 187 -188.

And in accordance with this ecological aspect, do authors expect a stronger/faster loss of MAAs in the wild, i.e. by climatic changes due to winter/summer season or humid/dry conditions at the time of sampling (as compared to stable herbarium conditions; lines 194-195)?

As long as we do not know the realized function(s) of MAAs in the Dermatocarpon thallus and the mechanisms by which MAA-concentrations are regulated in this lichen we can only speculate on the answer to this question.

However, based on the results from those samples which were taken from rocks which were submerged at the time of collection we can firmly reject a possible influence of rainfall on MAA-concentrations.

As outlined above, it is still unknown how MAAs are kept or replenished in the lichen thalli of a species like Dermatocarpon luridum var luridum which is inundated in running water for considerable amounts of time but still keeps up high MAA levels when sampled during a period of inundation.

Seasonal patterns of MAA contents are possible but should be studied with fresh material from the same populations under well monitored environmental conditions or – ideally – in the lab. For the herbarium collections used for our study the month of the collection is not given in all labels, particularly for most of the older collections. In more recently collected material we did not find a pattern which could indicate a seasonal change in MAA content. Dermatocarpon luridum is a semi-aquatic lichen, a potential role of MAAs as UV-screen is possible but may not be the only function. MAAs are also known to be involved in osmoregulation and as anti-oxidants, both roles may be of importance for coping with changes between total inundation and temporary dry conditions as they are typical for the splash water zone where the species is particularly common. Experimental approaches with fresh material are needed to answer this question.

Line 83: “… They are characterized by a unique strong absorption peak in the 310-360 nm UV range…”. However, only data for 310 nm are given in fig 2. What about the range/other wavelengths of 320-360?

A characteristic UV absorbance feature of MAAs is they exhibit only one symetrical and strong peak with a lmax between 310 and 360 nm. For mycosporines with a cyclohexanone ring: generally the lmax is around 310 nm, with a cyclohexenimine ring: the lmax is around 320 nm or 330 nm or 340 nm or 360 nm, it depends on the chemical structure.

We modified the sentence:

Line 83 :“They are characterized by a unique strong absorption peak in the 310-360 nm UV range with a lmax around 310 nm for MAAs having a cyclohexenone ring.

Fig. 2: the units on x and y axes are too small and therefore difficult to read; please, include a legend.

We added a legend as:

Figure 2. HILIC-HPLC chromatogram of the aqueous extract from a sample of Dermatocarpon luridum var. luridum (voucher barcode JB/001/07/2012) recorded at 310 nm and inset UV-spectra of mycosporines 1(Rt = 5.07 min) and 2 (Rt = 5.71 min) (IS: Internal Standard added to D. luridum extract, x-axis: 0-16 min, y-axis: arbitrary units of absorbance)

Fig. 3: please, comment on the varying content of glutamicol (in the results section). In the figure legend, it should be (blue circles) instead of triangles for glutaminol and the lichen species name should be in italic.

The taxon name is now in italics.

The symbols for mycosporine glutaminol are now consistently triangles (with blue filling if the measurements yielded concentrations > 0 mg/g, empty for specimens where no Mycosporine Glutaminol was detected).

- We completed the sentence:

line 157 : Mycosporine glutamicol contents varied between 4.56 and 34.89 mg/g of dried extract, corresponding to 0.15-1.12 mg/g of dried lichen (Figure 3), the four highest concentrations being recorded for samples collected in years 2006, 1933, 1886 and 1864.“

Discussion line 208: Are there differences in the (historical) use of herbarium sheets (paper, capsules, envelopes, etc.) in the two Natural History Collections in London and in Rennes? Have the authors compared samples of similar age out of the two difference Collections?

Yes we have compared collections of similar age from both collections but have not found significant differences in the pattern of MAA contents (see table in supplementary material).

Acidic paper is a typical side effect of industrial mass production of paper based on wood-pulp, starting in the mid 19th century and it continues to some extend as a problem to the present day. Explicitly “acid free” paper has been branded and became available for use in curatorial practice (again) at least since the 1980s at the NHM (UK). We do not have information on the quality of the paper used in the herbarium at Rennes University but based on visual inspection it appears to be distinctly discoloured in older collections, and therefore may have had a higher acidity compared to the British capsules and sheets.

However, there is also experimental evidence that various organic acids (incl. the volatile formic acid) are produced from natural aging of paper under ambient temperature conditions by oxidation of carbohydrate fragments such as the cellulose. This process also happens in papers which are branded as “acid free” or even alkaline paper types (Shahani & Harrisson 2002). We believe that this process (in addition to volatile substances released from lower quality acidic papers in older collections) is a plausible explanation for the equal decay of Mycosporine glutaminol in archival quality capsules as in capsules prepared from regular “office quality” paper. We have expanded the discussion on this topic accordingly and added the relevant reference.

Line 106: What is the meaning of the small "1" after Rennes?

There are two entities for Rennes Universities, University of Rennes 1 (= Sciences) and University of Rennes 2 (= Literature). We belong to the University of Rennes 1

Reviewer 3 Report

molecules-454125

Mycosporine-like amino acids (MAAs) in time series 2 of lichen specimens from Natural History Collections.

It is a paper  that makes a contribution to the area of natural products, however, it needs to be revised and improved in some of its sections

Some suggestions are given below.

1-In the section 4.2. Sample preparation

Number of samples processed should be indicated.

The authors writte “the yield of the extraction is about 3.2%.”

How many times the extraction was repeated? I suppose that experiments should be at least repeated in a triple mode.

2- Section 4.3. Method validation

The standards are usually purchased from a chemical company. Or the compounds are isolated and used as standards, which is this case. So the paragraph I think should be rewritten.

“Standards of mycosporine glutaminol (1) and mycosporine glutamicol (2) were isolated from D.  luridum [26].”

“Mycosporine glutaminol (1) and mycosporine glutamicol (2) isolated from D. luridum [26] were used as standards”

The spectroscopic methods for the structural determination of the compounds and how they established the purity of the compounds, should be mentioned in the text

3. Section 2.2. Analysis of lichen collections Figure 3.

This section is too preliminary.

The journal regularly publishes papers on the quantification of compounds from different origins.

A table with the results of the quantification should be included, according to the standards of the journal . Please look at recent publications.

Usually, data are given as mean±S.D.  (n=?)

Number of determinations in the tests should be mentioned in materials and methods.

3 repeated measurements of one stock or 3 single measurements of 3 different stocks or 3 repeated measurements for each of 3 different stocks?

After making the changes, the work could be considered for publication.

Author Response

1-In the section 4.2. Sample preparation

Number of samples processed should be indicated.

The number of samples is 48 as mentioned in Materials and Methods (line 252)

The authors writte “the yield of the extraction is about 3.2%.”

How many times the extraction was repeated? I suppose that experiments should be at least repeated in a triple mode.

We agree the best extraction procedure is to repeat in a triple mode, on a grinded material. However we conducted preliminary experiments in the lab (ref 19) demonstrating extraction yields were not really improved by repeating the extraction on grinded material with regard to a passive extraction of two hours with a large ratio of water and some shakings. So this simplest method, avoiding spills at the most, was preferred and applied in the same way in order to have a comparable analysis between the 48 samples. The main objective of this study is not the dosage accuracy but the estimation of the MAA’s content in long time series of herbarium specimens.

So we only corrected the term “grinded” which was a mistake:

Line 264 : “15 mg of uncrushed lichen material, …“

 2- Section 4.3. Method validation

The standards are usually purchased from a chemical company. Or the compounds are isolated and used as standards, which is this case. So the paragraph I think should be rewritten.

“Standards of mycosporine glutaminol (1) and mycosporine glutamicol (2) were isolated from D.  luridum [26].”

“Mycosporine glutaminol (1) and mycosporine glutamicol (2) isolated from D. luridum [26] were used as standards”

The spectroscopic methods for the structural determination of the compounds and how they established the purity of the compounds, should be mentioned in the text

The sentence line 271 was modified:

Line 271: “Mycosporine glutaminol (1) and mycosporine glutamicol (2) isolated from D. luridum [26] were used as standards. Their structures were determined by ESI+-HRMS, NMR and IR spectra as already published as SI in ref [26].”

 3. Section 2.2. Analysis of lichen collections Figure 3.

This section is too preliminary.

The journal regularly publishes papers on the quantification of compounds from different origins.

A table with the results of the quantification should be included, according to the standards of the journal . Please look at recent publications.

This table was submitted and is accessible as supplementary material.

Usually, data are given as mean±S.D.  (n=?)

Standard deviation was provided with the data in the submitted table for the supplementary material. Data are given as mean±S.D.  (n=2)

Number of determinations in the tests should be mentioned in materials and methods.

3 repeated measurements of one stock or 3 single measurements of 3 different stocks or 3 repeated measurements for each of 3 different stocks?

In this present work 2 repeated measurements of one stock sample solution were analyzed by HPLC. The sentence was modified accordingly lines 267-269:

Lines 267-269:“Crude aq. extracts were then dissolved in water and IS solution to give a final concentration at 1 mg/mL. Two repeated measurements of each sample solution were analyzed by HPLC.

Reviewer 4 Report

The current "Mycosporine-like amino acids (MAAs) in time series of lichen specimens from Natural History Collections" was organized well, and the findings are important for the application of Natural History Collections, as well as some contribution to other researchers related to MAAs. This paper fits the aim and scope of "Molecules" journal. 

The following comments should be modified as follows: 

1. The authors should provide the more mechanism explanations on why only mycosporine-glutaminol diminished within less than a decade. Authors provided good discussion at line 200-204. But, in Fig. 3, if this hypothesis is true, amounts of mycosporine-glutamicol should be increased when mycosporine-glutaminol was diminished.

2. In figure 3, please put the line, which show the average value of MAAs amounts, especially on mycosporine-glycol. This may help the understanding of readers.

Author Response

1. The authors should provide the more mechanism explanations on why only mycosporine-glutaminol diminished within less than a decade. Authors provided good discussion at line 200-204. But, in Fig. 3, if this hypothesis is true, amounts of mycosporine-glutamicol should be increased when mycosporine-glutaminol was diminished.

In lines 216-218 we have now taken into account the results from an overlooked paper by Shahani & Harrison (2002) which provide experimental evidence for the production of volatile organic acids under ambient storage conditions even for so-called “archival quality paper”.

The variation of Mycosporine-glutamicol concentrations is too large to expect a statistically robust effect of a possible addition from the conversion of Mycosporine glutaminol to the Mycosporine-glutamicol amount. Instead in lines 203-204 we have described the set-up for a robust test which will give a definite answer to the question if a conversion from Mycosporine glutamicol to Mycosporine glutaminol occurs.

2. In figure 3, please put the line, which show the average value of MAAs amounts, especially on mycosporine-glycol. This may help the understanding of readers.

To plot a line for average values across time would make sense if the different data points in the scatter plot represented true replicas of an experiment or sampling from thalli of the same location at a single moment of time. In figure 3 however each data represents already the average of two measurements from the same herbarium sample.

We believe that the creation of a line through the averages of these already averaged data points across time would only be correct if the samples represented collections from the same thallus at the same spot.

The composition of our data however was dictated by the distribution of the available samples. We have selected samples from the same overall region (Europe north of the Alps), but the locality information for most of the older collections is far too unprecise to conclude that collections from the same time are really from the same spot. Therefore we believe that creating a graphic generalization beyond the scatter plot would not add real benefit to the reader but rather risks the suggestion of a visual over-analysis. Our data set does deliver evidence for obvious differences in the trends of preservation for the two different MAAs in our model lichen, but due to the erratic pattern of collection sites and times the data cannot be easily compared to the results obtained from a controlled experiment and we believe that this should also be expressed by the graphic representation of our results.

Round 2

Reviewer 1 Report

LC/MS is definitely more sensitive than UV. If the compounds did not ionize well in the positive mode, the authors should try negative mode.

Author Response

Please find our response to reviewer report 1 in the attached PDF-file, including the results of an additional experiment to compare LC/MS vs. UV for the quantificatioin of the MAAs involved in our study.

Reviewer 3 Report

The authors have successfully made the suggested changes. The paper should be accepted in the present form.

Author Response

We appreciate the positive response from reviewer 3 to our revised version of the manuscript.